# A Transformer-Based Object Detector with Coarse-Fine Crossing Representations

**Zhishan Li**[1,2]  **Ying Nie**[2]  **Kai Han**[2]*  **Jianyuan Guo**[2]  **Lei Xie**[1]  **Yunhe Wang**[2]*

[1]College of Control Science and Engineering, Zhejiang University
[2]Huawei Noah's Ark Lab
`{zhishanli, lxie}@zju.edu.cn, {ying.nie, kai.han, yunhe.wang}@huawei.com`

## Abstract

Transformer-based object detectors have shown competitive performance recently. Compared with convolutional neural networks limited by the relatively small receptive fields, the advantage of transformer for visual tasks is the capacity to perceive long-range dependencies among all image patches, while the deficiency is that the local fine-grained information is not fully excavated. In this paper, we introduce the **C**oarse-grained and **F**ine-grained crossing representations to build an efficient **D**etection **T**ransformer (CFDT). Specifically, we propose a local-global cross fusion module to establish the connection between local fine-grained features and global coarse-grained features. Besides, we propose a coarse-fine aware neck which enables detection tokens to interact with both coarse-grained and fine-grained features. Furthermore, an efficient feature integration module is presented for fusing multi-scale representations from different stages. Experimental results on the COCO dataset demonstrate the effectiveness of the proposed method. For instance, our CFDT achieves 48.1 AP with 173G FLOPs, which possesses higher accuracy and less computation compared with the state-of-the-art transformer-based detector ViDT. Code will be available at `https://gitee.com/mindspore/models/tree/master/research/cv/CFDT`.

## 1   Introduction

Object detection is a fundamental task in the field of computer vision[1, 2]. The former mainstream architectures for object detection are mostly based on convolutional neural networks (CNNs)[3, 4, 5, 6, 7]. With the pioneering work of transformer [8] from natural language processing [9, 10] into object detection by DETR [11], its variants [12, 13, 14] show competitive detection performance [15], which can be attributed to the strong long-range dependency capturing ability.

Modern CNN-based object detectors, such as Faster-RCNN [16], YoloV3 [17], FCOS [18], and EfficientDet [19], can be divided into three components: backbone, neck and head. With the development of transformer in vision tasks, there are two common manners to deploy transformer for object detection. One is to replace the CNN-based backbones with transformer variants in object detectors. For example, some recently proposed transformer architectures like Swin Transformer [20], PVT [21, 22] and CMT [23] are utilized as backbone in the Mask-RCNN [24] or RetinaNet [25] detection frameworks. However, this manner heavily relies on the original detection frameworks, while anchor generation and post-processing with non-maximum suppression [26] are still indispensable. In this way, the role of transformer is just the backbone for feature extraction. The other manner is to replace the neck part with transformer [11], which discards the post-processing and anchor setting in conventional detection frameworks. Such a method still requires CNN to extract semantic information from images.

---

*Corresponding author

36th Conference on Neural Information Processing Systems (NeurIPS 2022).

The typical representatives of this approach are DETR [11] and its variants, such as Deformable-DETR [12], Efficient DETR [13], and DAB-DETR [27]. Carion *et al.* propose DETR [11] to firstly combine CNN and transformer to build an end-to-end detector. In DETR, ResNet [4] is used as the backbone for extracting features, and transformer is proposed to integrate the relations between learnable object queries and intermediate image features. However, there are two limitations for DETR. The first is the redundant computation brought by the encoder-decoder architecture of neck. The other is the slow convergence speed, which requires 500 epochs for training. Inspired by deformable convolution networks [28], Zhu *et al.* propose Deformable DETR [12], which replaces the original multi-head attention with deformable attention module. Besides, it aggregates multi-scale features in different stages of backbone, which is effective for object detection. With the multi-scale deformable attention module, Deformable DETR greatly exceeds DETR in both accuracy and training speed. In order to simplify the complexity of the transformer-based detection framework, Fang *et al.* proposes YoloS[47], which realizes fast object detection with a simple structure. To fully dig the potential of transformer in object detection task, Song *et al.* construct an efficient and effective fully transformer-based (both backbone and neck are transformer-based architectures) object detector called ViDT [14]. It adopts Swin-Transformer [20] as the backbone and reconfigures the attention module to support standalone object detection. In addition, it incorporates an encoder-free neck structure to further boost the detection performance without introducing too much computational burden. ViDT obtains the best AP and latency trade-off among existing transformer-based object detectors. Furthermore, some recently proposed transformer-based object detectors [29, 30, 31, 32, 33] show better performance than original CNN-based detectors.

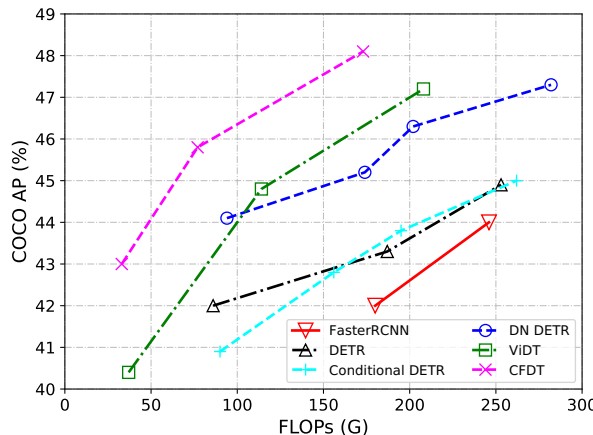

Figure 1: Performance comparison with other representative detectors on COCO 2017 val set. The FLOPs is calculated with a $800 \times 1333$ input image.

For transformer-based detection models like DETR [11] and ViDT[14], the typical strategy is to perform long-range attention on the divided feature patches. Compared with convolutional neural networks limited by a relatively small receptive field scale, the main advantage of transformer-based models is the capacity to perceive long-range dependencies among all image patches. However, the rich spatial information inside these divided patches is rarely considered by previous models. Take the general object detection benchmark as an example[1], there are objects with various sizes. Fine-grained representations can help to recognize multi-scale and irregular objects. For the previous transformer-based detectors, excessive pursuit of global feature representations yet paying less attention to local representations limits them for multi-scale perception. Therefore, it is crucial for object detectors to capture and fuse both fine-grained features inside the image patches and the global coarse-grained features to better detect objects with different scales.

In this paper, we propose to fully leverage both global **C**oarse-grained and local **F**ine-grained features to build an efficient **D**etection **T**ransformer (CFDT) with transformer backbone and transformer neck. In the backbone, we maintain both coarse-grained and fine-grained features and introduce a lightweight **L**ocal-**G**lobal **C**ross **F**usion (LGCF) module. In this way, a fully bidirectional cross fusion between local fine-grained and global coarse-grained information is carried out in each stage. In terms of neck, we propose **C**oarse-**F**ine **A**ware **N**eck (CFAN) which allows detection tokens to make attention-based interaction with fine-grained representations firstly, and then perform further interaction with coarse-grained representations. Finally, a lightweight bottom-up feature integration algorithm called **E**fficient **M**ultiscale **F**eature **I**ntergration (EMFI) is designed for enriching high resolution feature maps in the early stages. The extensive experiments demonstrate the effectiveness of the proposed method. As shown in Figure 1, our CFDT detectors obtain the best AP and FLOPs trade-off among existing transformer-based object detectors.

## 2 Preliminaries

In this section, we briefly revisit the fine-grained representations in vision transformers and the transformer-based detection frameworks.

### 2.1 Fine-grained Representations in Vision Transformers

Transformer-based models are recently applied in visual tasks. In general, they divide the original images into $N$ patches for capturing long-range dependencies between these $N$ patches [34, 35, 36, 37, 38]. However, such a framework destroys the internal relationship and ignores the fine-grained representations inside each patch. Han *et al.* [39] propose a Transformer iN Transformer (TNT) architecture that constructs not only the global connection among outer patches, but also the inner communication inside each patch. The outer patches describe global coarse-grained features while the inner patches represent local fine-grained information. Without loss of generality, denote $\mathcal{F}_O^{l-1}$ and $\mathcal{F}_I^{l-1}$ as the outer patches and inner patches input to the $l$-th stage, respectively. Correspondingly, denote $\mathcal{F}_O^l$ and $\mathcal{F}_I^l$ as the outer patches and inner patches output by the $l$-th stage, respectively. Then, the basic TNT block can be formulated as follows:

$$\mathcal{F}_I^l = \mathcal{F}_I^{l-1} + MLP(LN(\mathcal{F}_I^{l-1} + MSA(LN(\mathcal{F}_I^{l-1})))), \tag{1}$$

$$\mathcal{F}_O^l = \mathcal{F}_O^{l-1} + MLP(LN(\mathcal{F}_O^{l-1} + MSA(LN(\mathcal{F}_O^{l-1} + FC(\mathcal{F}_I^l))))), \tag{2}$$

where $MLP$, $LN$ and $MSA$ represent Multi-Layer Perceptron, Layer Normalization [40] and Multi-head Self-Attention, respectively. $FC$ represents the linear projection layer. In a word, Eq. 1 represents the inner transformer and Eq. 2 represents the outer transformer with inner attention. With TNT block, each outer patch can not only obtain the long-range dependency with other outer patches, but also integrate its corresponding finer-grained inner representations.

PyramidTNT [41] is the improved version of TNT, which introduces pyramid architecture and convolutional stem. With the relatively small amount of computation, PyramidTNT achieves a higher accuracy on ImageNet dataset [42]. Besides, the pyramid architecture is more suitable as the backbone of dense prediction tasks, such as object detection and instance segmentation. That is, with PyramidTNT as the backbone, the multi-scale inner fine-grained features and outer coarse-grained features can be easily obtained for the downstream tasks.

### 2.2 Detection Transformers

**DETR.** DETR utilizes ResNet as the backbone to extract features. In the neck part, it first adopts $6\times$ transformer blocks to perform self-attention on features, and then adopts another $6\times$ transformer blocks to perform cross attention between object queries and features. After the transformer-based neck, the final classification and regression results are predicted directly through detection heads. For the selection of training samples, DETR constructs the matching cost matrix between object queries and ground truths, and uses the Hungarian algorithm to efficiently calculate the optimal assignment [43]. DETR is an end-to-end framework, which does not need anchor boxes and non-maximum suppression.

**Deformable DETR.** There are still two deficiencies for DETR, including slow convergence and relatively poor detection performance for small objects. Deformable DETR proposes a deformable attention module, which attends to a small set of key sampling points around a reference. Besides, the deformable attention module can be naturally extended to aggregate multi-scale features, which is effective for object detection. Compared with DETR, Deformable DETR only needs 50 epochs to converge and greatly improve the detection performance of small objects.

**ViDT.** Compared with DETR and Deformable DETR, ViDT is a fully transformer-based detector. For the backbone, ViDT employs Swin-Transformer rather than ResNet. To fully utilize the transformer-based backbone, det tokens and patches share the same attention weights. In the last stage, ViDT constructs a reconfigured attention module to make cross attention between det tokens and patches. As for the neck, ViDT only retains the decoder part of Deformable DETR and its architecture is computationally efficient. Compared with other transformer-based detectors, ViDT obtains the best AP and latency trade-off.

# 3  Approach

In this section, we describe the proposed modules in detail. Firstly, we present the local-global cross fusion module to improve the backbone for object detection. Then, we illustrate the coarse-fine aware neck to further make det tokens interact with inner and outer patches. Finally, a lightweight bottom-up feature integration algorithm is introduced.

## 3.1  Local-Global Cross Fusion

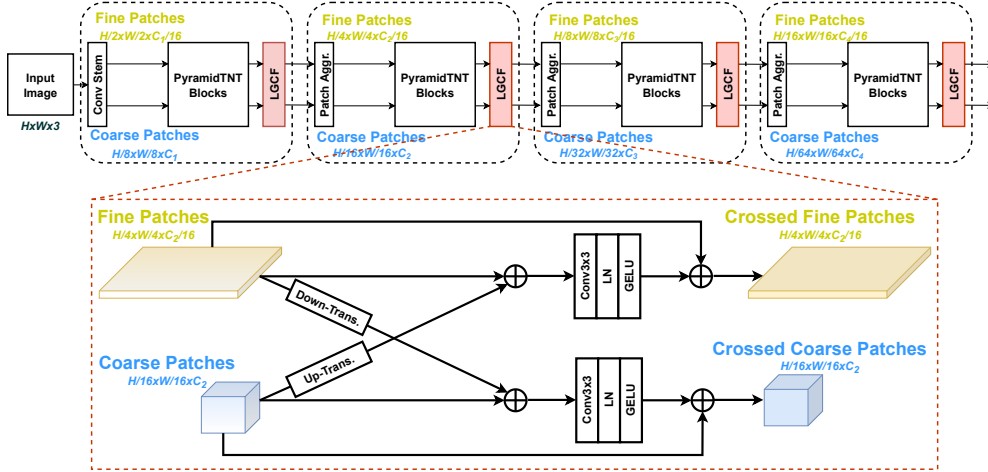

Figure 2: Illustration of LGCF embedded in the backbone of PyramidTNT in CFDT. Patch Aggr. represents patches aggregation, which is used to merge and reduce patches. Down-Trans. and Up-Trans. indicate spatial down-transform and up-transform of patches, respectively.

In TNT blocks or PyramidTNT blocks, from Eq. 2, although the outer patches can acquire the information of fine-grained inner representations, the restricted receptive field of inner patches and the unidirectional "Inner to Outer" strategy limit the diversity of inner features. Besides, the inner patches corresponding to each outer patch can only make self-attention in a fixed $4 \times 4$ region. As a result, they have no connection with inner patches that belong to other outer patches. Therefore, a **L**ocal-**G**lobal **C**ross **F**usion (LGCF) module is proposed here, as shown in Figure 2. LGCF can be divided into two sub-modules, including **L**ocal **C**ross **F**usion (LCF) and **G**lobal **C**ross **F**usion (GCF).

Formally, given a 2D image $\mathcal{I} \in \mathbb{R}^{H \times W \times 3}$, the outer coarse-grained patches and inner fine-grained patches output by the $l$-th stage are denoted as $\mathcal{F}_O^l \in \mathbb{R}^{\frac{H}{2^{l+2}} \times \frac{W}{2^{l+2}} \times C_l}$ and $\mathcal{F}_I^l \in \mathbb{R}^{\frac{H}{2^l} \times \frac{W}{2^l} \times \frac{C_l}{16}}$, respectively. For the inner patches, LCF is proposed to fuse them with long-range dependency to expand the receptive field. Since the relationship between outer patches is global, LCF brings the perception of global information to the original inner patches which only attach fixed $4 \times 4$ internal self-attention. The process of LCF can be formulated as follows:

$$Cross_I^l = \mathcal{F}_I^l + Upsample(Conv_{1\times1}(\mathcal{F}_O^l))))), \tag{3}$$

$$\mathcal{F}_I^l = \mathcal{F}_I^l + GELU(LN(Conv_{3\times3}(Cross_I^l))). \tag{4}$$

Eq. 3 describes the cross operation from outer to inner. In the equation, $Conv_{1\times1}$ and $Upsample$ represent convolution with kernel size $1 \times 1$ and upsampling with bilinear interpolation respectively. The point convolution is adopted to keep the features' channel of outer patches consistent with inner patches, and the upsampling operation is adopted to expand the spatial scale of outer patches to 16 times. The transformed outer patches are in line with inner patches both in spatial dimension and channel dimension. $Cross_I^l$ is the summation of inner patches and transformed outer patches. Then, we use the combination of "Convolution-Normalization-Activation" to further fuse the crossed features, as described in Eq. 4. $Conv_{3\times3}$ represents the convolution operation with $3 \times 3$ kernel size. $LN$ and $GELU$ represent Layer Normalization [40] and Gaussian Error Linear Unit activation [44], respectively.

Although the outer patches acquire fine-grained inner representations with original PyramidTNT blocks, the fusion method of simple flattening and addition ignores spatial information. Similar to LCF, we propose GCF to integrate inner features into outer patches. This process can be formulated as follows:

$$Cross_O^l = \mathcal{F}_O^l + Conv_{4\times4}(\mathcal{F}_I^l), \tag{5}$$

$$\mathcal{F}_O^l = \mathcal{F}_O^l + GELU(LN(Conv_{3\times3}(Cross_O^l))). \tag{6}$$

Eq. 5 represents the cross operation from inner patches to outer patches. Different from the combination of point convolution and upsample in Eq. 3, we directly use a $Conv_{4\times4}$ to get a transformed $\mathcal{F}_I^l$ consistent with $\mathcal{F}_O^l$ in both spatial shape and channel. In Eq. 6, we also use the combination of "Convolution-Normalization-Activation" to further fuse the crossed features.

With this module, bidirectional cross fusion is carried out between local features and global features after each stage. The local representations integrate more global information and the original global coarse-grained representations fuse the fine-grained information. Experimental results show that the proposed LGCF can greatly improve the detection performance.

### 3.2 Coarse-Fine Aware Neck

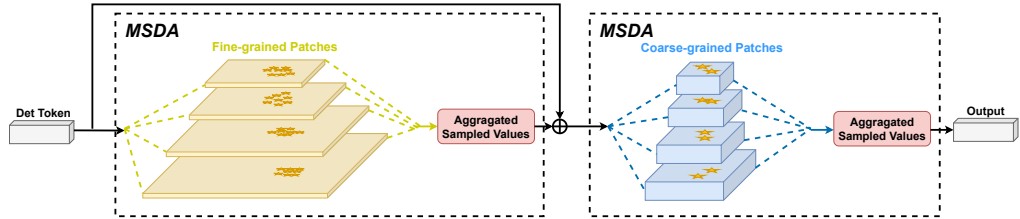

Figure 3: Illustration of the coarse-fine aware neck in CFDT. MSDA is the abbreviation of Multi-Scale Deformable Attention. The small brown star indicates sampling point.

No matter DETR or ViDT, the neck part always performs the cross attention operation between det tokens and global long-range dependency features. In object detection task, there are usually targets with different scales in different positions. Therefore, it is necessary to pay attention to multi-scale and multi-source useful features. Here, we propose a **C**oarse-**F**ine **A**ware **N**eck (CFAN) module, which allows det tokens to interact with not only coarse-grained outer patches but also with fine-grained inner patches. In practice, we perform Multi-Scale Deformable Attention (MSDA) [12] between det tokens and local fine-grained features firstly, then between det tokens and global coarse-grained features, as illustrated in Figure 3. The cross attention between det tokens and inner patches can be formulated as follows:

$$MSDA(Q_{det}, \{\mathcal{F}_I^l\}_{l=1}^L) = \sum_{m=1}^{M_I} W_m \left[ \sum_{l=1}^L \sum_{k=1}^{K_I} A_{mlk} \cdot W_m' \mathcal{F}_I^l(\phi_l(p) + \Delta p_{mlk}) \right], \tag{7}$$

where

$$Q_{det} = Q_{det} + MSDA(Q_{det}, \{\mathcal{F}_I^l\}_{l=1}^L), \tag{8}$$

where $Q_{det}$ represents the det tokens. Eq. 7 represents the MSDA process between det tokens and inner patches. $M_I$ indicates the attention head and $K_I$ is the total number of sampled keys in inner patches. Besides, $\phi_l(p)$ represents the reference point in the $l$-th stage features, while $\Delta p_{mlk}$ represents the corresponding sampling offset for the next deformable attention operation. $A_{mlk}$ is the attention weights of the $K$-th sampling contents. $W_m$ and $W_m'$ are the projection matrices in multi-head attention operation.

After the deformable cross attention of det tokens and inner patches, we first combine the $MSDA(Q_{det}, \{\mathcal{F}_I^l\}_{l=1}^L)$ to $Q_{det}$ and then interact with $\mathcal{F}_O$. This process can be formulated as follows:

$$MSDA(Q_{det}, \{\mathcal{F}_O^l\}_{l=1}^L) = \sum_{m=1}^{M_O} W_m \left[ \sum_{l=1}^L \sum_{k=1}^{K_O} A_{mlk} \cdot W_m' \mathcal{F}_O^l(\phi_l(p) + \Delta p_{mlk}) \right], \tag{9}$$

where $M_O$ indices the attention head and $K_O$ is the total number of sampled keys in outer patches. Consistent with the original MSDA in Deformable DETR [12], we set the default values of $M_I$ and $M_O$ to 8 in the following experiments.

### 3.3 Efficient Multi-scale Feature Intergration

For an input image $\mathcal{I} \in \mathbb{R}^{H \times W \times 3}$, the shapes of output features in four stages in our backbone can be set as $\frac{H}{8} \times \frac{W}{8}$, $\frac{H}{16} \times \frac{W}{16}$, $\frac{H}{32} \times \frac{W}{32}$, $\frac{H}{64} \times \frac{W}{64}$, respectively. In contrast, the shapes of output features in four stages in Swin-Transformer or ResNet should be $\frac{H}{4} \times \frac{W}{4}$, $\frac{H}{8} \times \frac{W}{8}$, $\frac{H}{16} \times \frac{W}{16}$, $\frac{H}{32} \times \frac{W}{32}$, respectively. In ViDT or Deformable DETR, when utilizing Swin-Transformer or ResNet as backbone, the output features of the first stage is not used by the neck due to the insufficient useful information. The downsampled features of the last stage are usually taken

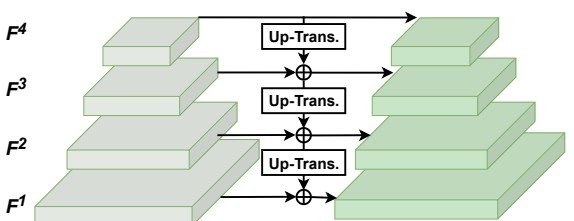

Figure 4: Illustration of the efficient multi-scale feature intergration module in CFDT. $F^l$ represents the output patches (inner or outer) in $l$-th stage.

as an additional features in their methods. In object detection task, the shapes of features have a great impact on the final performance. In general, it is more effective to use large-scale features for small object detection. Therefore, although the feature extraction capability of the first stage is limited, we still cannot ignore the output features of this stage.

To effectively utilize the output of the first stage, we design an **E**fficient **M**ultiscale **F**eature **I**ntegration (EMFI) module, as shown in Figure 4. We insert EMFI into the fine-grained and coarse-grained features respectively, which can enhance the feature representation of the early stages. This module can be formulated as follows:

$$\mathcal{F}_O^l = \mathcal{F}_O^l + Upsample(Conv_{1 \times 1}(\mathcal{F}_O^{l+1})), \tag{10}$$

$$\mathcal{F}_I^l = \mathcal{F}_I^l + Upsample(Conv_{1 \times 1}(\mathcal{F}_I^{l+1})). \tag{11}$$

The output of $l$-th stage is the summation of original features and transformed output of $(l+1)$-th stage. The transformer operation is consist of bilinear interpolation and point convolution, which can bring ignorable computational cost. Such a bottom-up structure enables the high-resolution features of the early stages to integrate the later features with low resolution but richer semantic information.

## 4 Experiments

In this section, we conduct extensive experiments. Thorough ablation studies are also provided in this section.

### 4.1 Dataset and Implementation Details

We conduct experiments on Microsoft COCO 2017 benchmark [1]. Following the usual practice, 118K images are used for training, 5K images for testing. In addition, we follow the training strategy provided in ViDT [14], including AdamW [45] with the initial learning rate of $1 \times 10^{-4}$, training with multi-scale input sizes and the total training epoch is set as 50. We use PyramidTNT series models pretrained on ImageNet-1K as the initial backbones of detectors. The results are reported over three backbones: PyramidTNT-Tiny (P-Tiny), PyramidTNT-Small (P-Small), and PyramidTNT-Medium (P-Medium). Considering the existence of Batch Normalization [46] in the stem of backbone and the batch size of training object detection network is much smaller than that in image classification, we freeze all parameters of Batch Normalization layers in the training process. Besides, the training batch size per card of P-Tiny and P-Small is set as 2, while that of P-Medium is 1. For evaluation, Average Precision (AP) is calculated on COCO, and the floating point operations (FLOPs) is calculated under a $800 \times 1333$ input image.

Table 1: Comparisions of CFDT with other transformer-based detectors on COCO 2017 val set.

| Model | Backbone | $AP$ | $AP_{50}$ | $AP_{75}$ | $AP_S$ | $AP_M$ | $AP_L$ | FLOPs (G) |
|---|---|---|---|---|---|---|---|---|
| *FLOPs (G) Range: 10∼50* | | | | | | | | |
| YOLOS[47] | DeiT-Tiny | 30.4 | 48.6 | 31.1 | 12.4 | 31.8 | 48.2 | 21 |
| ViDT*[14] | Swin-Nano | 40.4 | 59.6 | 43.3 | 23.2 | 42.5 | 55.8 | 37 |
| **CFDT** * | **P-Tiny** | **43.0** | 62.5 | 45.8 | 23.9 | 45.7 | 59.7 | 33 |
| *FLOPs (G) Range: 50∼150* | | | | | | | | |
| DETR[11] | ResNet-50 | 42.0 | 62.4 | 44.2 | 20.5 | 45.8 | 61.1 | 86 |
| Conditional DETR[48] | ResNet-50 | 40.9 | 61.8 | 43.3 | 20.8 | 44.6 | 59.2 | 90 |
| DAB DETR[27] | ResNet-50 | 42.2 | 63.1 | 44.7 | 21.5 | 45.7 | 60.3 | 94 |
| UP DETR[49] | ResNet-50 | 42.8 | 63.0 | 45.3 | 20.8 | 47.1 | 61.7 | 86 |
| DN DETR[50] | ResNet-50 | 44.1 | 64.4 | 46.7 | 22.9 | 48.0 | 63.4 | 94 |
| SAM DETR[51] | ResNet-50 | 41.8 | 63.2 | 43.9 | 22.1 | 45.9 | 60.9 | 100 |
| ViDT*[14] | Swin-Tiny | 44.8 | 64.5 | 48.7 | 25.9 | 47.6 | 62.1 | 114 |
| **CFDT*** | **P-Small** | **45.8** | 65.3 | 49.2 | 25.9 | 48.5 | 63.6 | 77 |
| *FLOPs(G) Range: 150∼300* | | | | | | | | |
| DETR[11] | ResNet-101 | 43.5 | 63.8 | 46.4 | 21.9 | 48.0 | 61.8 | 152 |
| DETR[11] | DC5-ResNet-50 | 43.3 | 63.1 | 45.9 | 22.5 | 47.3 | 61.1 | 187 |
| DETR[11] | DC5-ResNet-101 | 44.9 | 64.7 | 47.7 | 23.7 | 49.5 | 62.3 | 253 |
| Efficient DETR[13] | ResNet-50 | 45.1 | 63.1 | 49.1 | 28.3 | 48.4 | 59.0 | 210 |
| Efficient DETR[13] | ResNet-101 | 45.7 | 64.1 | 49.5 | 28.2 | 49.1 | 60.2 | 289 |
| Conditional DETR[48] | ResNet-101 | 42.8 | 63.7 | 46.0 | 21.7 | 46.6 | 60.9 | 156 |
| Conditional DETR[48] | DC5-ResNet-50 | 43.8 | 64.4 | 46.7 | 24.0 | 47.6 | 60.7 | 195 |
| Conditional DETR[48] | DC5-ResNet-101 | 45.0 | 65.5 | 48.4 | 26.1 | 48.9 | 62.8 | 262 |
| SMCA[52] | ResNet-50 | 45.6 | 65.5 | 49.1 | 25.9 | 49.3 | 62.6 | 152 |
| SMCA[52] | ResNet-101 | 46.3 | 66.6 | 50.2 | 27.2 | 50.5 | 63.2 | 218 |
| DAB DETR[27] | ResNet-101 | 43.5 | 63.9 | 46.6 | 23.6 | 47.3 | 61.5 | 174 |
| DAB DETR[27] | DC5-ResNet-50 | 44.5 | 65.1 | 47.7 | 25.3 | 48.2 | 62.3 | 202 |
| DAB DETR[27] | DC5-ResNet-101 | 45.8 | 65.9 | 49.3 | 27.0 | 49.8 | 63.8 | 282 |
| DN DETR[50] | ResNet-101 | 45.2 | 65.5 | 48.3 | 24.1 | 49.1 | 65.1 | 174 |
| DN DETR[50] | DC5-ResNet-50 | 46.3 | 66.4 | 49.7 | 26.7 | 50.0 | 64.3 | 202 |
| DN DETR[50] | DC5-ResNet-101 | 47.3 | 67.5 | 50.8 | 28.6 | 51.5 | 65.0 | 282 |
| Deformable DETR[12] | ResNet-50 | 45.4 | 64.7 | 49.0 | 26.8 | 48.3 | 61.7 | 173 |
| SAM DETR[51] | DC5-ResNet-50 | 45.0 | 65.4 | 47.9 | 26.2 | 49.0 | 63.3 | 210 |
| YOLOS[47] | DeiT-Small | 36.1 | 55.7 | 37.6 | 15.6 | 38.4 | 55.3 | 194 |
| ViDT*[14] | Swin-Small | 47.4 | 67.7 | 51.2 | 30.4 | 50.7 | 64.6 | 208 |
| **CFDT*** | **P-Medium** | **48.1** | 67.8 | 51.8 | 28.1 | 50.9 | 66.4 | 173 |

∗ denotes that backbone and neck are both transformer-based architecture.

## 4.2   Main Results

We compare our method with latest transformer-based detectors, including DETR[11], SMCA[52], UP DETR[49], Efficient DETR[13], Conditional DETR[48], DAB DETR[27], DN DETR[50], SAM DETR[51], YOLOS[47] and ViDT[14], as shown in Table 1. For fair comparison, all the results do not utilize the strategy of multi-scale test.

**Compare with tiny detectors.** YOLOS[47] is a canonical ViT architecture for object detection. Although it has a small computational cost, the neck-free design withholds the YOLOS from obtaining high performance, our CFDT achieves +12.7 AP compared to the Deit-tiny based YOLOS. When compared to the recently proposed lightweight ViDT[14], our CFDT outperforms it by +2.6 AP with fewer FLOPs. More specifically, the backbones of ViDT and CFDT attain similar results on ImageNet (74.9 of Swin-Nano v.s. 75.2 of P-Tiny), and the superiority in COCO further demonstrates the improvements brought by our proposed LGCF, CFAN, and EMFI.

**Compare with small detectors.** We further compare our P-small based CFDT with Swin-tiny based ViDT and several variants of ResNet-50 based DETR. For example, DN DETR[50] accelerates DETR training by introducing query denoising. Our CFDT outperforms it by +1.7 AP with far less FLOPs (-17G), and we still exceed the ViDT by +1.0 AP, and the FLOPs is significantly reduced by 37G.

**Compare with medium detectors.** For the backbone with P-Medium, our CFDT achieves 48.1 AP with 173G FLOPs. In terms of AP, the detectors close to our method are DN DETR with DC5-ResNet-101 and ViDT with Swin-Small. Compared with them, the FLOPs of our method is lower than these detectors by 109G and 35G, respectively. Besides, our method still reaches a better detection performance.

When compared to those detectors with ResNet as the backbone, transformer-based models like ViDT and CFDT show the better trade-off between accuracy and computational cost (higher AP and fewer FLOPs). This also reveals that the detectors which consist of transformer-based backbone and transformer-based neck possess great potential for efficient object detection.

### 4.3 Ablations

**Comparison between PyramidTNT and Swin-Transformer.** In order to illustrate that our detection performance is not due to the backbone replacement of PyramidTNT, we firstly show the comparison of PyramidTNT and Swin-Transformer in Table 2. Although the Top-1 accuracy of PyramidTNT series on ImageNet is slightly higher than that of the corresponding sized Swin-Transformer models, AP of directly taking PyramidTNT-Small or PyramidTNT-Medium as backbone of ViDT is significantly lower than that of Swin-Transformer. For further analysis, compared with PyramidTNT, Swin-Transformer utilizes the "Shift Window" to obtain multi-scale features, which is effective for object detection. So for transformer-based object detection, it is not good enough to directly deploy PyramidTNT as backbone.

Table 2: Comparisons between Swin-Transformer and PyramidTNT.

| Models | ImageNet (Top-1) | COCO (AP)$^\star$ |
|---|---|---|
| Swin-Nano | 74.9 | 40.4 |
| PyramidTNT-Tiny | **75.2** | **40.8** |
| Swin-Tiny | 81.3 | **44.8** |
| PyramidTNT-Small | **82.0** | 43.4 |
| Swin-Small | 83.0 | **47.4** |
| PyramidTNT-Medium | **83.5** | 44.3 |

$\star$ AP is obtained by taking ViDT as detection framework and the corresponding models set as backbone.

**Local-Global Cross Fusion.** We analyze the impact of different elements in LGCF, as shown in Table 3. The baseline of our method with backbone of P-Tiny is 40.8 AP. After introducing the LCF to fuse the outer coarse-grained patches into fine-grained inner patches, AP is improved to 41.3. Correspondingly, it is more effective to embed GCF in the detector. After the introduction of GCF, we achieve an AP improvement of 1.2 compared with baseline. For further analysis, the improvement of GCF is higher than LCF, but it is also accompanied by a higher amount of calculation. The reason is that the channels of outer patches are 16 times that of inner. So the convolution operation in GCF brings more computation. Finally, we combine the two cross fusion strategies together and achieve an improvement of 1.4 AP compared with baseline.

Table 3: Analysis of Local-Global Cross Fusion module with backbone of P-Tiny.

| Backbone | LCF | GCF | AP | $\Delta_{AP}$ | FLOPs (G) |
|---|---|---|---|---|---|
| | | | 40.8 | - | 27.8 |
| | ✓ | | 41.3 | ↑0.5 | 28.5 |
| P-Tiny | | ✓ | 42.0 | ↑1.2 | 30.8 |
| | ✓ | ✓ | **42.2** | **↑1.4** | 31.5 |

**Sampling Points of CFAN.** We conduct an ablation experiment on the sampled points number $K_I$ of inner fine-grained patches. The baseline is the P-Tiny backbone with LGCF whose AP is 42.2. For outer patches, we set $K_O$ as 4, which is consistent with Deformable DETR and ViDT. Considering that one outer patch corresponds to 16 inner patches, so $K_I$ is set to an integer multiple of 16. The result is shown in Table 4. When $K_I$ is set to 0, there is no cross attention between det tokens and fine-grained patches in the neck part. At this case, det tokens directly interact with outer global coarse-grained patches through Multi-Scale Deformable Cross Attention modules. When $K_I$ equals to 32, AP increases to 42.3 with det tokens interacting with 32 sampled keys of inner patches. The

best performance is $K_I$ set to 64, which obtains 42.6 AP and 0.4 higher than baseline. At this case, $K_I$ equals 64 inner points that 4 sampled outer keys correspond to.

Table 4: Effect of sampled keys number $K_I$ of inner fine-grained patches in P-Tiny.

| $K_I$ | 0 | 16 | 32 | 48 | 64 | 80 |
|---|---|---|---|---|---|---|
| AP | 42.2 | 42.2 | 42.3 | 42.5 | **42.6** | 42.4 |

**Complete Component Analysis.** We analyze all components in CFDT, and the detailed result is shown in Table 5. For Tiny model as backbone, we find that the LGCF, CFAN and EMFI improve the AP of 1.4, 0.4, 0.4 respectively. Combining all the proposed modules together, our method achieve 43.0 AP, which is 2.2 AP higher than baseline. In addition, the increase of computation brought by the introduction of these three modules is acceptable. Compared with ViDT based on Swin-Nano (40.4 AP, 37G FLOPs), our method has less computation cost, but possesses 2.6 AP higher than Swin-Nano. For other backbones of our method, although the proposed modules bring in a little computation cost, the FLOPs of CFDT is still less than that of other models. Compared with other transformer-based detectors, our method achieve a higher AP.

Table 5: Analysis of all components in CFDT.

| Backbone | LGCF | CFAN | EMFI | AP | $\Delta_{AP}$ | FLOPs (G) |
|---|---|---|---|---|---|---|
| P-Tiny | | | | 40.8 | - | 27.8 |
| | ✓ | | | 42.2 | ↑1.4 | 31.5 |
| | ✓ | ✓ | | 42.6 | ↑1.8 | 33.0 |
| | ✓ | ✓ | ✓ | **43.0** | **↑2.2** | 33.2 |
| P-Small | | | | 43.4 | - | 65.1 |
| | ✓ | | | 45.0 | ↑1.6 | 74.6 |
| | ✓ | ✓ | | 45.3 | ↑1.9 | 76.1 |
| | ✓ | ✓ | ✓ | **45.8** | **↑2.4** | 76.5 |
| P-Medium | | | | 44.3 | - | 150.0 |
| | ✓ | | | 46.5 | ↑2.2 | 170.8 |
| | ✓ | ✓ | | 47.2 | ↑2.9 | 172.3 |
| | ✓ | ✓ | ✓ | **48.1** | **↑3.8** | 173.2 |

### 4.4 Visualization of sampling points in CFAN

To better understand the Coarse-Fine Aware Neck, we randomly select three images and visualize the sampling points in both coarse-grained patches and fine-grained patches, as shown in Figure 5. From the visualization, it is apparently that the sampling points on coarse-grained patches are more concentrated in a certain area on foreground, while the sampling points on fine-grained patches are more widely distributed. However, many sampling points on fine-grained patches are assigned to the background area. Therefore, it is necessary to combine them together to capture the effective features for better detection.

## 5 Conclusion

In this paper, we propose an efficient object detector called CFDT with coarse-grained and fine-grained cross representations. In order to further improve the performance of detector, we propose local-global cross fusion module, coarse-fine aware neck and efficient multi-scale feature intergration strategy. Compared with the state-of-the-art transformer-based detector ViDT, the combination of our approach achieves better detection performance with less computation cost. Among other transformer-based detectors, our method obtains a better trade-off between AP and FLOPs. Experimental results demonstrate the effectiveness of the proposed method. For future research, we hope to transfer this idea to more transformer-based models, so as to improve the performance of various visual tasks with transformer.

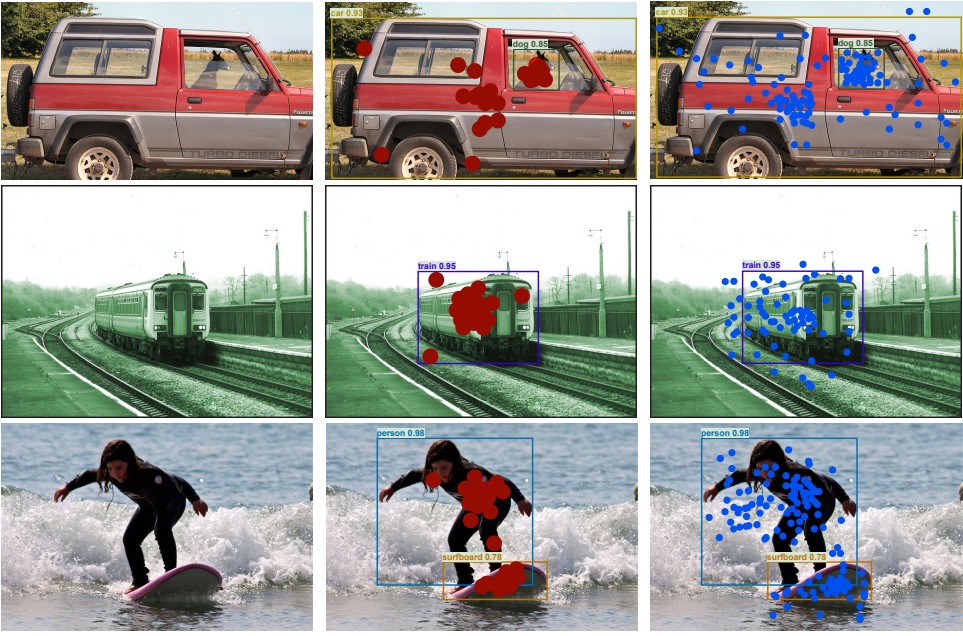

Figure 5: Illustration of the sampling points in CFAN. We transform the sampling points location of corresponding prediction boxes to original images. The bigger red points indicate the sampling points on coarse-grained patches and the smaller blue points represent the sampling points on fine-grained patches.

## Acknowledgement

We gratefully acknowledge the support of MindSpore [53], CANN(Compute Architecture for Neural Networks) and Ascend AI Processor used for this research. This research is supported by National Natural Science Foundation of P.R. China (NSFC: 62073286).

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
