# Supplementary Material: A Transformer-Based Object Detector with Coarse-Fine Crossing Representations

**Zhishan Li**[1,2]  **Ying Nie**[2]  **Kai Han**[2*]  **Jianyuan Guo**[2]  **Lei Xie**[1]  **Yunhe Wang**[2*]

[1]College of Control Science and Engineering, Zhejiang University
[2]Huawei Noah's Ark Lab
{zhishanli, lxie}@zju.edu.cn, {ying.nie, kai.han, yunhe.wang}@huawei.com

## A   Appendix

### A.1   More details about the overall detector

The overall architecture of CFDT is shown in Figure 1. We elaborate on the details in terms of backbone and neck.

**Backbone.** The base backbone is consistent with the network illustrated in the section of 3.1 Local-Global Cross Fusion. The other details that we need to pay attention are the connection between det tokens and image patches. As shown by the red dotted lines in Figure 1, we use 100 det tokens as the additional input to perform self attention in the backbone. In this process, all det tokens share the same weights $(W_Q, W_K, W_V)$ with coarse-grained patches, as YOLOS or ViDT. Therefore, the embedding dimension of det tokens is equal to that of the outer coarse patches. Due to the feature channels of patches are increased by patches aggregation operation $(C_1 \rightarrow C_2 \rightarrow C_3 \rightarrow C_4)$ at the end of each stage, we duplicate the channels of each det token to keep them consistent with coarse patches. In addition, we make cross attention between det tokens and coarse patches in the last stage, as ViDT does. For the last stage multi-head attention mechanism, the query contents are from det tokens, and the key and value contents are from the concatenation of det tokens and coarse patches. Therefore, in the last stage, det tokens directly interact with the image patches.

**Coarse-Fine Aware Neck.** Due to the different channel dimension of different stage outputs, we use projection layers to set the embedding dimension of coarse-grained patches and fine-grained patches to 256 and 16 respectively. The det tokens dimension is also set as 256. The Coarse-Fine Aware Neck is a decoder-only modules, and there are 6 decoder layers in this neck. For each decoder layer, there are two Multi-Scale Deformable Cross-Attention interacting with Fine-grained patches and Coarse-grained patches respectively.

### A.2   More comparison results with other detectors

Besides the comparison with other transformer-based detectors, we also compare our CFDT with RetinaNet $1\times$ and RetinaNet $3\times$ using recently proposed transformer as backbone. The detailed results are shown in Table 1. From the table, our method achieves the best 48.1 AP, while the FLOPs is only 173G. In the future object detection frameworks, transformer-based detectors maybe have the potential to become the mainstream models. In the meanwhile, some small modules consisted of CNN can be used to further make up for the disadvantages of transformer.

---

*Corresponding author

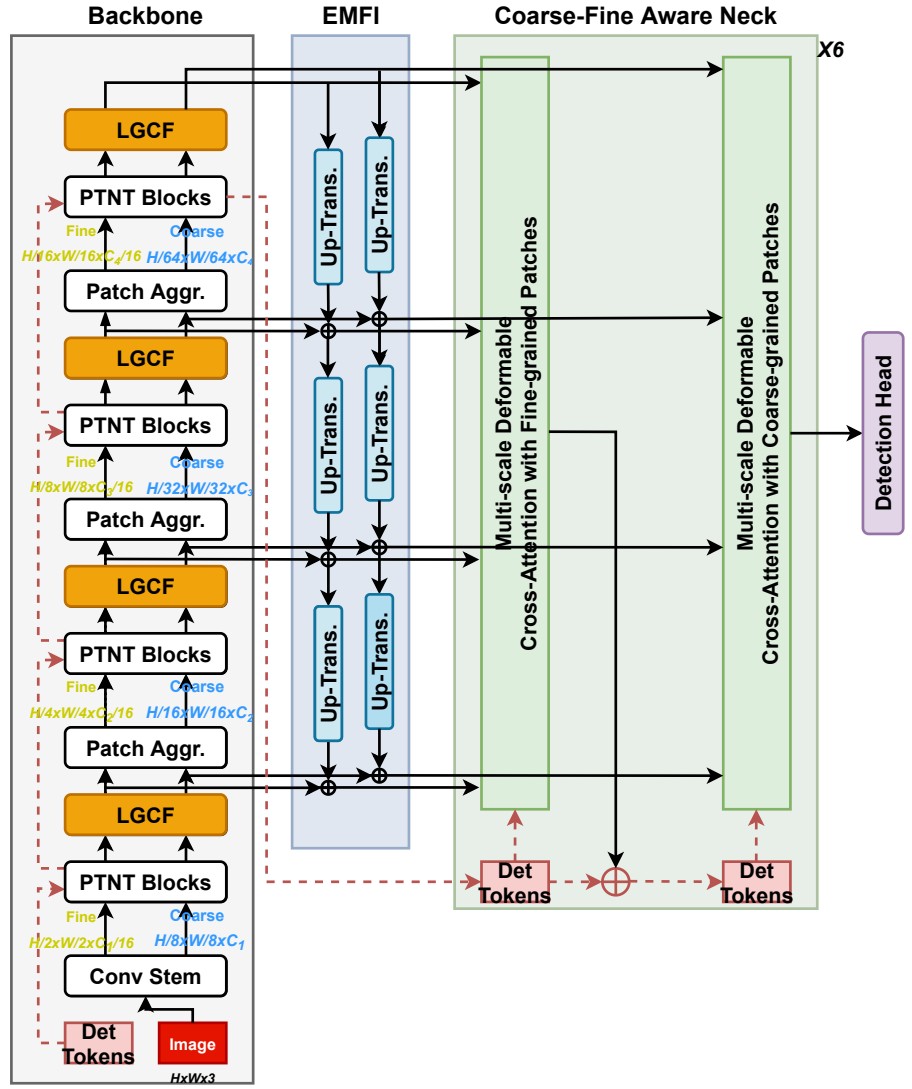

Figure 1: Illustration of the overall architecture in CFDT. PTNT Blocks is the abbreviation of PyramidTNT Blocks. The red dotted line represents the forward propagation of det tokens.

### A.3 Coarse-Fine Crossing Representations in ViDT

To show the extensibility of our proposed methods in other Transformer-based detector, we migrate the LGCF and CFAN to ViDT with the backbone of Swin-Transformer. We insert the inner patches into Swin-Transformer, and introduce the LGCF to perform cross fusion between coarse-grained and fine-grained features. We also utilize CFAN to let det tokens interact with both types of representations.

In Swin-Transformer, the image patches are generated by the "PatchEmbed" operation. We regard the obtained image patches as outer patches and generate inner patches from the input image. We keep outer patches as before to extract features through Swin-Transformer blocks. For inner patches, we extract local fine-grained features through a new independent basic Transformer block (including Multi-Head Attention and MLP, *etc.* ) in each stage. At the end of each stage, we perform the mutual cross fusion between global coarse-grained features and local fine-grained features through LGCF. Compared with the original model, we add a basic transformer block and a LGCF module in each stage. Besides, we also use CFAN to let det tokens make cross-attention with both types of

Table 1: Comparison with RetinaNet $1\times$ and RetinaNet $3\times$ using transformer as backbone. All backbones are pretrained on ImageNet-1K. We omit models pretrained on larger-datasets (e.g., ImageNet-21K). The FLOPs (G) range is 150~300.

| Detection Framework | Backbone | $AP$ | $AP_{50}$ | $AP_{75}$ | $AP_S$ | $AP_M$ | $AP_L$ | FLOPs (G) |
|---|---|---|---|---|---|---|---|---|
| RetinaNet $1\times$ [1] | PVT-T[2] | 39.4 | 59.8 | 42.0 | 25.5 | 42.0 | 52.1 | 221 |
| | PVT-S[2] | 42.2 | 62.7 | 45.0 | 26.2 | 45.2 | 57.2 | 226 |
| | PVT-M[2] | 41.9 | 63.1 | 44.3 | 25.0 | 44.9 | 57.6 | 283 |
| | PVTv2-B0[3] | 37.2 | 57.2 | 39.5 | 23.1 | 40.4 | 49.7 | 177 |
| | PVTv2-B1[3] | 41.2 | 61.9 | 43.9 | 25.4 | 44.5 | 54.3 | 225 |
| | PVTv2-B2[3] | 44.6 | 65.6 | 47.6 | 27.4 | 48.8 | 58.6 | 290 |
| | MPViT-T[4] | 41.8 | 62.7 | 44.6 | 27.2 | 45.1 | 54.2 | 196 |
| | MPViT-XS[4] | 43.8 | 65.0 | 47.1 | 28.1 | 47.6 | 56.5 | 211 |
| | MPViT-S[4] | 45.7 | 57.3 | 48.8 | 28.7 | 49.7 | 59.2 | 248 |
| | Swin-T[5] | 42.0 | 63.0 | 44.7 | 26.6 | 45.8 | 55.7 | 245 |
| | Focal-T[6] | 43.7 | 65.2 | 46.7 | 28.6 | 47.4 | 56.9 | 265 |
| | Twins-SVT-S[7] | 42.3 | 63.4 | 45.2 | 26.0 | 45.5 | 56.5 | 209 |
| | Twins-PCPVT-S[7] | 43.0 | 64.1 | 46.0 | 27.5 | 46.3 | 57.3 | 226 |
| | Shunted-S[8] | 45.4 | 65.9 | 49.2 | 28.7 | 49.3 | 60.0 | - |
| | CMT-S[9] | 44.3 | 65.5 | 47.5 | 27.1 | 48.3 | 59.1 | 231 |
| RetinaNet $3\times$ [1] | PVT-T[2] | 39.4 | 59.8 | 42.0 | 25.5 | 42.0 | 52.1 | 221 |
| | PVT-S[2] | 42.2 | 62.7 | 45.0 | 26.2 | 45.2 | 57.2 | 226 |
| | PVT-M[2] | 43.2 | 63.8 | 46.1 | 27.3 | 46.3 | 59.9 | 283 |
| | MPViT-T[4] | 44.4 | 65.5 | 47.4 | 29.9 | 48.3 | 56.1 | 196 |
| | MPViT-XS[4] | 46.1 | 67.4 | 49.3 | 31.4 | 50.2 | 58.4 | 211 |
| | MPViT-S[4] | 47.6 | 68.7 | 51.3 | 32.1 | 51.9 | 61.2 | 248 |
| | Swin-T[5] | 45.0 | 65.9 | 48.4 | 29.7 | 48.9 | 58.1 | 245 |
| | Focal-T[6] | 45.5 | 66.3 | 48.8 | 31.2 | 49.2 | 58.7 | 265 |
| | Twins-SVT-S[7] | 45.6 | 67.1 | 48.6 | 29.8 | 49.3 | 60.0 | 209 |
| | Twins-PCPVT-S[7] | 45.2 | 66.5 | 48.6 | 30.0 | 48.8 | 58.9 | 226 |
| | Shunted-S[8] | 46.4 | 66.7 | 50.4 | 31.0 | 51.0 | 60.8 | - |
| | CMT-S[9] | 46.9 | 67.1 | 50.5 | 30.4 | 49.8 | 61.0 | 231 |
| **CFDT** | **P-Medium** | **48.1** | **67.8** | **51.8** | **28.1** | **50.9** | **66.4** | **173** |

representations. We utilize Swin-Nano as the base backbone, and the experimental results are as follows.

Table 2: Analysis of Coarse-Fine Crossing Representations in ViDT.

| Backbone | LGCF | CFAN | AP | $\Delta_{AP}$ | FLOPs (G) |
|---|---|---|---|---|---|
| Swin-Nano | | | 40.4 | - | 37 |
| | ✓ | | 42.3 | ↑1.9 | 43 |
| | ✓ | ✓ | 42.7 | ↑2.3 | 45 |

From Table 2, it is obvious that the combination of LGCF and CFAN greatly improves 2.3 AP for ViDT. The trend of AP changes is consistent with CFDT.

## A.4 Limitations and societal impacts

The main limitation of CFDT is that it still divides the whole framework into several sub modules, including backbone and neck. Actually, backbone is mainly used to extract features, while neck is mainly used to make cross attention between det tokens and features. Because CFDT is a transformer-based detector, it is more promising to combine the two parts into one module. In other words, a single model can perform image features extraction and interacting between det tokens and image patches. We hope to propose such a detector in the near future.

As for the societal impacts, because CFDT possesses the characteristics of high accuracy and low computation, it may be deployed to monitoring and other scenarios. If these are obtained by criminals, there might be social risks of information security disclosure.