# OpenReview forum: "A Transformer-Based Object Detector with Coarse-Fine Crossing Representations"
_NeurIPS.cc/2022/Conference — NeurIPS 2022 Accept_

### Official Review · Reviewer_6R9f · 2022-07-08

**Rating:** 5
**Confidence:** 5
**Soundness:** 3 good
**Presentation:** 3 good
**Contribution:** 2 fair

**Summary:**

This paper studies the topic of 'Fully Transformer-based Object Detector' (except the stem layer, do not use CNN layers for operations). Existing Transformers mainly rely on global interaction between outer patches, but this work proposes a way of integrating attention of inner patches, capturing local & fine-grained information; (1) LGCF: mixing fine & coarse patches with interacting them, (2) FCAN: alternating cross-attention btw <det x fine-grained patch> and <det x coarse-grained patches>, and (3) EMFI: using the first block patches efficiently. The proposed architecture shows a better AP and Speed trade-off compared with other compared detectors. To support their justification, the authors provide some ablation studies on the proposed three components.

**Questions:**

1) Could you explain why $AP_s$ drops compared with ViDT (Swin-small)?
2) I feel hard to know what components make the architecture faster than others. I understand your design is simple and effective but there is no explanation why it is faster than another efficient mechanism (e.g., local attention in Swin - ViDT).
3) Your architecture seems to be coupled with PyramidTNT. This could be a limitation (low extendability). Do you think your three components can be combined with other backbones as well? If possible, doing and comparing them would be good to prove that the performance improvement is not from the backbone.



**Limitations:**

Refer to the strengths and weaknesses above.

**Strengths And Weaknesses:**

There are both strengths and weaknesses I found while reading the paper.

### Strength
1) Integrating the idea of 'local + global information' into Transformer-based object detection is reasonable because improving AP on varying-size objects (especially, small-size) is a major challenge in this domain.
2) The proposed three components (blocks) look simple and effective when combined. Therefore, they did not harm the inference speed of the detector.
3) Experimental results are not promising but reasonable improvements.

### Weakness
1) ***Performance drop for small-size objects***. According to the experiments, $AP_s$ drops from 30.4 to 28.1 compared to ViDT, although more fine-grained information was integrated. I can't understand why this happens. Intuitively, the proposed ideas of fusing more fine-grained information should improve $AP_s$.
2) ***Incomplete Analysis on Backbones***. In Section 4.3, there is a comparison between PyramidTNT and Swin-Transformer. I think this analysis is not complete to show that the improvement did not come from the replacement. First, ViDT is not designed to be with PyramidTNT. Second, only accuracy and AP measures are reported without 'FLOPS or FPS'. To sum up these, I think a more suitable comparison is combining Swin-Transformer with your three components. The performance trends with ViDT could not tell us their impact on your architecture.
3) ***Weak Novelty***. The proposed approach mostly relies on existing works. For local- fine-grained information, PyramidTNT is adopted as the backbone, and the proposed three components are very simple and straightforward; this could be an advantage in practice, but academically looks like an incremental paper.

---

> ### Author Response · Authors · 2022-08-02
> **Responses to Reviewer 6R9f  (Part 3/3)**
>
> **Q2: I feel hard to know what components make the architecture faster than others. I understand your design is simple and effective but there is no explanation why it is faster than another efficient mechanism (e.g., local attention in Swin - ViDT).**
>
> **A-Q2:** Thanks for the helpful comment.
>
> First of all, compared with ResNet and Swin-Transformer, PyramidTNT is a relatively lightweight model. In the upper **A-W1&Q1**, we illustrate that the scale of feature maps output in the four stages is smaller than that of Swin-Transformer. It is also such a characteristic that the number of patches in each stage of PyramidTNT is 1/4 of that of Swin-Transformer, so its calculation amount is greatly reduced compared with Swin-Transformer. In addition, the number and feature dimension of transformer blocks are also smaller than that of Swin-Transformer. The FLOPs of ViDT with the backbone of Swin-Nano is 37G. After we replace it with PyramidTNT-Tiny, the FLOPs are reduced to 28G. The FLOPs of ViDT with the backbone of Swin-Tiny is 114G. After we replace it with PyramidTNT-Small, the FLOPs are reduced to 65G.
>
> Then, our designed components are lightweight. LGCF accounts for the most computation in the proposed modules. For the backbone, since we only introduce LGCF at the end of each stage, there are only four LGCF modules added to the backbone. The detailed changes of FLOPs are shown in the table below. With the backbone of PyramidTNT-Tiny, the FLOPs increase of the proposed three components are 3.7G, 1.5G, and 0.2G, respectively.
>
> | Backbone  | LGCF | FCAN|EMFI | AP   | FLOPs(G) |
> |-|-|-|-|-|-|
> | PyramidTNT_Tiny|      |      | |40.8 | 27.8       |
> | PyramidTNT_Tiny|  ✓    |      | |42.2 | 31.5      |
> | PyramidTNT_Tiny|   ✓   |   ✓   | |42.6 | 33.0      |
> | PyramidTNT_Tiny|   ✓   |   ✓   | ✓|43.0 | 33.2       |
>
> With the above two factors,  our proposed FCDT is efficient compared with ViDT. Because our proposed modules have brought obvious $AP$ improvement with less calculation, this is an advantage of practical application.
>
> **Q3: Your architecture seems to be coupled with PyramidTNT. This could be a limitation (low extendability). Do you think your three components can be combined with other backbones as well? If possible, doing and comparing them would be good to prove that the performance improvement is not from the backbone.**
>
> **A-Q3:** Thanks for the helpful comment.
>
> The proposed components can be combined with other backbones as well. As the answer **A-W2**, we insert the inner patches into Swin-Transformer, and introduce the LGCF to perform cross fusion between coarse-grained and fine-grained features. We also utilize FCAN to let det tokens interact with both types of representations.  Since the purpose of EMFI is to make up for the gap between feature extraction ability of PyramidTNT and Swin-Transformer in $\frac{H}{8} \times \frac{W}{8} $ feature maps, it is not necessary to deploy EMFI to ViDT with the backbone of Swin-Transformer.
>
> In Swin-Transformer, the image patches are generated by the "PatchEmbed" operation. We regard the obtained image patches as outer patches and generate inner patches from the input image. We keep outer patches as before to extract features through Swin-Transformer blocks. For inner patches, we extract local fine-grained features through a new independent basic Transformer block(including Multi-Head Attention and MLP, etc) in each stage. At the end of each stage, we perform the mutual cross fusion between global coarse-grained features and local fine-grained features through LGCF.
>
> Compared with the original model, we add a basic transformer block and a LGCF module in each stage. Besides, we also use FCAN to let det tokens make cross-attention with both types of representations. We utilize Swin-Nano as the base backbone, and the experimental results are as follows:
>
> | Backbone  | LGCF | FCAN | AP   | FLOPs(G) |
> |-|-|-|-|-|
> | Swin_Nano |      |      | 40.4 | 37       |
> | Swin_Nano |  ✓    |      | 42.3 | 43       |
> | Swin_Nano |   ✓   |   ✓   | 42.7 | 45       |
>
>
> Compared with the baseline of 40.4 $AP$, the introduction of LGCF increases to 42.3 $AP$. With FCAN, the $AP$ further increases to 42.7.  This performance proves that our proposed components can be extended to another backbone. We show that our method is feasible for general Transformer-based detector to capture fine-coarse crossing representations.
>
>
>
> **Again, thank you very much for the kind efforts in evaluating and helping to improve the quality of our manuscript!**

---

> > ### Comment · Reviewer_6R9f · 2022-08-08
> > **Raised Score**
> >
> > I appreciate the authors' response. Now, it's clear why the proposed architecture is faster than Swin-based models. The reason mainly comes from the small-scale feature maps in Pyramid-TNT. Although it reduces the AP on small-size objects, the paper's proposed components help improve it without compromising inference speed. The provided results are good enough to show the contribution of each component. Therefore, I raise my score and recommend putting the new results and analysis in the paper.

---

> > > ### Author Response · Authors · 2022-08-08
> > > **Authors' response**
> > >
> > > Thank you for the support. We will add the new results and analysis in the paper.

---

> ### Author Response · Authors · 2022-08-02
> **Responses to Reviewer 6R9f  (Part 2/3)**
>
> **W2: Incomplete Analysis on Backbones. In Section 4.3, there is a comparison between PyramidTNT and Swin-Transformer. I think this analysis is not complete to show that the improvement did not come from the replacement. First, ViDT is not designed to be with PyramidTNT. Second, only accuracy and AP measures are reported without 'FLOPS or FPS'. To sum up these, I think a more suitable comparison is combining Swin-Transformer with your three components. The performance trends with ViDT could not tell us their impact on your architecture.**
>
> **A-W2:** Thanks for the helpful comment.
>
> The original intention of the comparison in Section 4.3 is to show that our FCDT achieves great performance, not because the backbone is just replaced by PyramidTNT, but to highlight the effectiveness of the three proposed components. If there is no such comparison, it might cause a misunderstanding that the main reason why we can achieve good results is utilizing PyramidTNT as backbone. In fact, directly replacing the backbone of ViDT with PyramidTNT cannot reach the detection performance that Swin-Transformer achieves.  On the other hand, it can be seen from Tab.2 in Section 4.3 that the effect of image classification is not necessarily positively correlated with the performance of object detection.  With such a comparison, we highlight that the better detection performance of FCDT mostly comes from the three modules that we propose.
>
> However, we realize that this is indeed an incomplete analysis of backbones because ViDT is not designed to be with PyramidTNT. Therefore, in order to prove the scalability of the proposed algorithm, we migrate the LGCF and FCAN to ViDT with the backbone of Swin-Transformer. Since the purpose of EMFI is to make up for the gap between feature extraction ability of PyramidTNT and Swin-Transformer in $\frac{H}{8} \times \frac{W}{8} $ feature maps, it is not necessary to deploy EMFI to ViDT with the backbone of Swin-Transformer. We insert the local fine-grained information into Swin-Nano and introduce the LGCF and FCAN. Experimental results are as follows. From the table, it is apparent that our LGCF and FCAN greatly improve 2.4 AP. The trend of AP changes is consistent with FCDT. Some specific details are shown in **A-Q3**.
> | Backbone  | LGCF | FCAN | AP   | FLOPs(G) |
> |-|-|-|-|-|
> | Swin_Nano |      |      | 40.4 | 37       |
> | Swin_Nano |  ✓    |      | 42.3 | 43       |
> | Swin_Nano |   ✓   |   ✓   | 42.7 | 45       |
>
>
>
> **W3: Weak Novelty. The proposed approach mostly relies on existing works. For local- fine-grained information, PyramidTNT is adopted as the backbone, and the proposed three components are very simple and straightforward; this could be an advantage in practice, but academically looks like an incremental paper.**
>
> **A-W3:** Thanks for the helpful comment.
>
> The advantage of transformer-based models is the capacity to perceive long-range dependencies among all image patches. Therefore, this is also the main reason why transformer-based detectors, such as DETR, YOLOS, ViDT, etc., can achieve great detection performance.  However, these models ignore the spatial local information within each patch.  There are useful fine-grained features inside each divided patch, which are rarely considered. Although the fine-grained inner representations have been proposed in TNT or PyramidTNT, the restricted receptive field of inner patches and the unidirectional "Inner to Outer" method limit the performance of inner representations. Besides, the previous "Inner to Outer" method is simply flattening the inner patches and adding to outer patches, which cannot fully exploit spatial information inside fine-grained features.  Therefore, we propose LGCF for mutual cross fusion between global coarse-grained features and local fine-grained features. Besides, we proposes FCAN to make full use of fine-grained features, which let det tokens make cross-attention with both fine and coarse grained representations.  Furthermore, we introduce EMFI to use the first stage outputs efficiently.
>
> The main contribution of this paper is to introduce the Fine-grained and Coarse-grained crossing representations. We hope to provide a new idea to optimize the detection transformer, which is to capture both local fine-grained features and global coarse-grained features.  In order to show that this is feasible, we introduce FCAN and LGCF to ViDT with the backbone of Swin-Transformer.  In future research, we hope that the proposed coarse-fine grained crossing strategies can inspire more Transformer-based models.

---

> ### Author Response · Authors · 2022-08-02
> **Responses to Reviewer 6R9f  (Part 1/3)**
>
> **We would like to express our sincere thanks for the constructive comments and suggestions from the reviewer.**
>
>
>
> **W1: Performance drop for small-size objects. According to the experiments,  $AP_s$ drops from 30.4 to 28.1 compared to ViDT, although more fine-grained information was integrated. I can't understand why this happens. Intuitively, the proposed ideas of fusing more fine-grained information should improve $AP_s$.**
>
> **Q1: Could you explain why $AP_s$ drops compared with ViDT (Swin-small)?**
>
> **A-W1&Q1:** Thanks for the helpful comments. Since these two questions are about $AP_s$, it is appropriate to answer them together.
>
> **The difference of backbones (Swin vs. PyramidTNT)**. Compared with ViDT with the backbone of Swin-Small, the proposed FCDT with the backbone of PyramidTNT-Medium achieves higher $AP$, but lower $AP_ S$. The main factor is the relatively weak feature representations output by the first stage blocks. Feature maps with larger size are more suitable for detecting small objects. For a 2D image with a scale of $H \times W $, with ResNet or Swin-Transformer as the backbone, the scale of feature maps in each stage is $\frac{H}{4} \times \frac{W}{4} $, $\frac{H}{8} \times \frac{W}{8} $, $\frac{H}{16} \times \frac{W}{16} $, $\frac{H}{32} \times \frac{W}{32} $. Feature maps of the last 3 stages are usually fed into Neck, and an additional feature map is generated by downsampling after the last stage. Therefore, taking ResNet or Swin-Transformer as the backbone, the scale of four output feature maps are $\frac{H}{8} \times \frac{W}{8} $, $\frac{H}{16} \times \frac{W}{16} $, $\frac{H}{32} \times \frac{W}{32} $,$\frac{H}{64} \times \frac{W}{64} $. These features are used in the Neck part to make cross-attention with det tokens. In PyramidTNT, the scale of feature maps in each stage is $\frac{H}{8} \times \frac{W}{8} $, $\frac{H}{16} \times \frac{W}{16} $, $\frac{H}{32} \times \frac{W}{32} $,$\frac{H}{64} \times \frac{W}{64} $. In object detection, the feature map scale has a great impact on detection performance, so we keep the outputs shape of our method consistent with Swin-Transformer or ResNet. Therefore, for the feature map with the same scale of $\frac{H}{8} \times \frac{W}{8} $, the feature map obtained by Swin-Transformer is generated by 2 stages of Swin-Transformer blocks, while ours is only 1 stage of Transformer block. So the feature representations of scale with $\frac{H}{8} \times \frac{W}{8} $ in our method are relatively weak. The detection performance of small objects for large backbone PyramidTNT-Medium is not as good as ViDT. In terms of PyramidTNT-Tiny and PyramidTNT-Small with relatively small complexity, we acquire higher $AP_ S$ than ViDT with the backbone of Swin-Transformer.
>
> **The performance gains of the proposed modules**. In fact, the proposed three components increased the $AP$, as well $AP_S$.  As shown in Tab.5 of the original paper, we reveal the analysis of complete components. The below table shows the improvement for $AP_S$ under the backbone of PyramidTNT-Medium.  It can be seen that compared with directly deploying PyramidTNT as the backbone, the introduction of Local-Global Cross Fusion (LGCF), Fine-Coarse Aware Neck (FCAN) and Efficient Multi-scale Feature Integration (EMFI) not only increases the $AP$, but also greatly improves the $AP_ S$ from 23.1 to 28.1. Therefore, our proposed methods are beneficial for the improvement of small object detection.
>
> | Backbone  | LGCF | FCAN|EMFI | AP   | APs |
> |-|-|-|-|-|-|
> | PyramidTNT_Medium|      |      | |44.3 | 23.1       |
> | PyramidTNT_Medium|  ✓    |      | |46.5 | 26.7      |
> | PyramidTNT_Medium|   ✓   |   ✓   | |47.2 | 27.0      |
> | PyramidTNT_Medium|   ✓   |   ✓   | ✓|48.1 | 28.1       |
>
>
> **Swin-Transformer takes into account the multi-scale effect.** With shifted windows, patches obtain features of different scales. This design has already achieved good detection performance. Besides, we introduce LGCF and FCAN to ViDT with the backbone of Swin-Nano, and the results are as follows:
> | Backbone  | LGCF | FCAN | AP   |APs |
> |-|-|-|-|-|
> | Swin_Nano |      |      | 40.4 | 23.2      |
> | Swin_Nano |  ✓    |      | 42.3 | 25.7       |
> | Swin_Nano |   ✓   |   ✓   | 42.7 | 25.9      |
>
> It is apparent that the introduction of LGCF and FCAN not only increases the AP, but also the detection performance for small objects. Some specific details about how to immigrate them to Swin-Transformer are shown in **A-Q3**.

---

### Official Review · Reviewer_8pQM · 2022-07-10

**Rating:** 7
**Confidence:** 4
**Soundness:** 3 good
**Presentation:** 3 good
**Contribution:** 3 good

**Summary:**

This paper introduces the Fine-grained and Coarse-grained crossing representations for building a Detection Transformer by using a local-global cross fusion module and Fine-Coarse Aware Neck.

**Questions:**

- Why are features of outer and features of inner different sizes in Line148?
- What happens when det tokens interacts with coarse-grained outer patches first and then with fine-grained inner patches in section3.2?
- Based on Eq. 8, my understanding is that Qdet and MSDA(Qdet, Fi) have a shortcut, but why Qdet and MSDA(Qdet, Fo) do not have shortcut?


**Limitations:**

The authors have discussed the limitations and potential negative societal impact of their work in A.4.

**Strengths And Weaknesses:**

Strengths:
+ The paper is well organized.
+ The performance boost seems good with a slight increase in Flops.

Weaknesses:
- The whole pipeline seems to be fine, but lacks deep insight. Most of the operations are feature aggregation between different levels, which has been deeply explored in object detection.
- Some of the statements in this paper are confusing. For example, how to understand “combine the MSDA(...) to Qdet and interact with Fo” in Line192.
- Writing errors. For example, “the lightweight bottom-up feature integration algorithm Efficient Multi-scale Feature Integration” should be “the lightweight bottom-up feature integration algorithm, i.e., Efficient Multi-scale Feature Integration” in Line138. The authors are advised to double-check for similar errors.

Overall, the simple and lightweight structure brings a huge performance boost is the reason I agree to accept this article.

---

> ### Author Response · Authors · 2022-08-02
> **Responses to Reviewer 8pQM  (Part 3/3)**
>
>
>
> **Q2: What happens when det tokens interacts with coarse-grained outer patches first and then with fine-grained inner patches in section3.2?**
>
> **Q3: Based on Eq. 8, my understanding is that Qdet and MSDA(Qdet, Fi) have a shortcut, but why Qdet and MSDA(Qdet, Fo) do not have shortcut?**
>
> **A-Q2&A-Q3:** Thanks for the helpful comments. Since these two questions are about the Neck, it is appropriate to answer them together.
>
> The main advantage of Transformer is the long-range global attention mechanism between each patch. And the internal information can be used as auxiliary features to improve fitting ability. Because each inner patch only performs attention with other 15 inner patches in a fixed region, we only need to use fewer transformer blocks to extract local fine-grained features compared with complex global representations. Taking PyramidTNT-Medium as an example, the number of transformer blocks used for global features in each stage is 2,8,6,2 respectively, while the number used for extracting local representations is 1,2,1,1. And the introduction of local representations does not increase too much computation burden. Therefore, the main representations are the global features, while the local fine-grained features are auxiliary. From the experimental results, the introduction of local fine-grained features is very effective for object detection.
>
> For Q2, with PyramidTNT-Tiny as the backbone, if we let det tokens interact with coarse-grained features first and then with fine-grained features, the AP decreases from 42.2 to 41.7. Oppositely, our proposed method with FCAN significantly improves the AP to 42.6. The reason is as follows. Transformer-based detectors such as YoloS and ViDT only consider global coarse-grained features, so their det tokens directly interact with outer patches. After making cross-attention with extracted features in the Neck part, the prediction results are acquired by putting the det tokens into prediction head (MLP). So the outputs of the Neck that can be directly connected with the prediction head are more important. Local fine-grained features are auxiliary to helps global coarse-grained features obtain local information, but they are not mainstream features. So we let det tokens interact with fine-grained inner patches first and then with coarse-grained outer patches.
>
> For Q3, the role of Neck is to let det tokens make cross-attention with feature maps. We utilize the Multi-Scale Deformable Attention (MSDA) to complete this cross-attention. In previous detectors, i.e., ViDT or Deformable-DETR, the output is $MSDA(Q_{det},\mathcal{F})$ and there is no shortcut from $Q_{det}$. In our method, the previous trial was to perform $MSDA(Q_{det},\mathcal{F}\_{I})$ first, then is another MSDA between $MSDA(Q_{det},\mathcal{F}\_{I})$ and $\mathcal{F}\_{O}$. However, the AP slightly decreased from 42.2 to 42.0 in this way. With the introduction of a shortcut to the MSDA between $Q_{det}$ and $\mathcal{F}\_{I}$, AP is significantly improved to 42.6. The reason is as follows. Det tokens share the same weights with outer patches in the backbone. For the fourth stage in the backbone, det tokens also perform cross-attention with outer patches. Hence, compared with inner patches, there is a close relation between det tokens and outer patches.  In terms of local fine-grained features as an auxiliary role, the interaction between det tokens and local features is also an assistant to increase the perception of det tokens. Therefore, the introduction of a shortcut is to make identity mapping to ensure that original det tokens can still fully interact with outer patches in the Neck. Since det tokens have been closely related to outer patches in the backbone, we directly perform MSDA between det tokens and global outer patches, as ViDT does.
>
>
> **Again, thank you very much for the kind efforts in evaluating and helping to improve the quality of our manuscript!**

---

> > ### Comment · Reviewer_8pQM · 2022-08-09
> > **Responses to Author**
> >
> > Thanks for your reply, which solves my question. I keep my original score.

---

> > > ### Author Response · Authors · 2022-08-09
> > > **Authors' response**
> > >
> > > Thank you very much for your recognition of our manuscript.

---

> ### Author Response · Authors · 2022-08-02
> **Responses to Reviewer 8pQM  (Part 2/3)**
>
>
> **W3: Writing errors. For example, “the lightweight bottom-up feature integration algorithm Efficient Multi-scale Feature Integration” should be “the lightweight bottom-up feature integration algorithm, i.e., Efficient Multi-scale Feature Integration” in Line138. The authors are advised to double-check for similar errors.**
>
> **A-W3:** Thanks for the helpful comment. It is indeed a writing error of original sentence “we introduce the lightweight bottom-up feature integration algorithm Efficient Multi-scale Feature Integration module.” We have changed it to “ we introduce the lightweight bottom-up feature integration algorithm, i.e., Efficient Multi-scale Feature Integration”. We have also checked the whole paper again, mainly to correct the typos and grammar errors.
>
> Some of these errors are modified as follows：
>
> In Line18, the original description is "**In the past decade,  models based on convolutional neural networks (CNNs) used to be the mainstream architecture for object detection tasks.**",  the changed version is "**The former mainstream architectures for object detection are mostly based on convolutional neural networks (CNNs).**"
>
> In Line25, the original description is "can be divided into three **parts**: backbone, neck and head",  the changed version is "can be divided into three **components**: backbone, neck, and head."
>
> In Line26, the original description is "With the development of transformer **used for** vision tasks", the changed version is "With the development of transformer **in** vision tasks".
>
> In Line38, the original description is "In DETR, ResNet is used as backbone for extracting features and transformer is proposed to integrate the relations between learnable object queries and extracted image features", the changed version is "In DETR, ResNet is used as **the backbone** for extracting features and transformer is proposed to integrate the relations between learnable object queries and **intermediate** image features".
>
> In Line46, the original description is "The other is the slow **convergence**", the changed version is "The other is the slow **convergence speed**".
>
> In Line61, the original description is "it incorporates an encoder-free neck structure to **boost** the detection performance without **much increase in computational load**", the changed version is "it incorporates an encoder-free neck structure to **further** **boost** the detection performance without **introducing too much computational burden**".
>
> In Line70, the original description is "**Considering the general object detection task**", the changed version is "**Take the general object detection benchmark as an example**".
>
> In Line91, the original description is "In this section, we briefly revisit the fine-grained representations in transformer and **detection transformer frameworks**", the changed version is "In this section, we briefly revisit the fine-grained representations in transformer and **transformer-based detection frameworks**".
>
>
> **Q1: Why are features of outer and features of inner different sizes in Line148?**
>
> **A-Q1:** For a 2D image with the shape of $H\times W \times C$, we divide it into $\frac{H}{4} \times \frac{W}{4} $ outer patches. The shape of each outer patch is $4\times 4 \times C$, so the length of each outer patch in the process is $16C$. At this time, the shape of outer patches can be written as [$\frac{H}{4} \times \frac{W}{4} $,$16C$]. In addition, each outer patch is also composed of $4\times 4 $ pixels. We regard its internal pixels as inner patches, and the length of each inner patch is $C$. Each inner patch only performs attention with other 15 inner patches in a fixed region. At this time, the shape of inner patches can be written as [$H\times W$, $C$]. The inner patches and outer patches can be considered as two feature maps with different scales and channel numbers. Therefore, for the $l$ stage feature maps, we use $\mathcal{F}\_{O}^{l} \in \mathbb{R} ^{\frac{H}{2^{l+2}}\times \frac{W}{2^{l+2}}\times C_{l}}$ and $\mathcal{F}\_{I}^{l} \in \mathbb{R} ^{\frac{H}{2^{l}}\times \frac{W}{2^{l}}\times \frac{C_{l}}{16}}$ to represent outer coarse-grained patches and inner fine-grained patches, respectively.

---

> ### Author Response · Authors · 2022-08-02
> **Responses to Reviewer 8pQM  (Part 1/3)**
>
> **We would like to express our sincere thanks for the constructive comments and suggestions from the reviewer.**
>
>
>
> **W1: The whole pipeline seems to be fine, but lacks deep insight. Most of the operations are feature aggregation between different levels, which has been deeply explored in object detection.**
>
> **A-W1:** Thanks for the helpful comment.
>
> Compared with CNN, the reason why Transformer can achieve great performance in visual tasks is the typical strategy to perform global long-range attention on the divided image patches. Therefore, this is also the main reason why transformer-based detectors, such as DETR, YOLOS, ViDT, etc., achieve great detection performance.
>
> However, these models ignore the spatial local information inside each patch.  There are useful fine-grained features inside each divided patch, which are rarely considered. The main contribution of this paper is to introduce the Fine-grained and Coarse-grained crossing representations, and propose a cross-fusion method to capture both local and global information in the meanwhile. Through the proposed Local-Global Cross Fusion (LGCF), our model achieves a significant improvement in accuracy.  Besides, we put forward Fine-Aware Neck (FCAN) to make full use of fine-grained features, which let det tokens make cross-attention with both fine and coarse grained representations.  Furthermore, we put forward Efficient Multi-scale Feature Integration (EMFI) to use the first stage feature map efficiently. The other transformer-based detectors, such as DETR, ViDT, etc., pursue better global information but do not consider the local fine-grained features. Through fine-coarse crossing representations, our proposed FCDT possesses high accuracy and less computation. Therefore, it is promising to introduce local fine-grained representations and explore better crossing methods between coarse and fine grained features to enhance the detection performance of Transformer-based detectors.
>
> **W2: Some of the statements in this paper are confusing. For example, how to understand “combine the MSDA(...) to Qdet and interact with Fo” in Line192.**
>
> **A-W2:** Thanks for the helpful comment. The original word “combine” is inappropriate. This statement in Line 192 should change to “After the deformable cross attention of det tokens and inner patches, we add $MSDA(Q\_{det},\mathcal{F}\_{I})$ to $Q\_{det}$, and then make further interact with $\mathcal{F}\_{O}$”. We have reexamined the whole paper again and found some unclear descriptions. We will rewrite those sentences to make them not confusing. Some of them are modified as follows：
>
> In Line27, the original description is "One is to **replace the backbone with transformer variants in CNN-based object detectors**", the changed version is "One is to **replace the CNN-based backbones with transformer variants in object detectors**".
>
> In Line68, the original description is "However, **these models ignore the spatial local information within each patch. There are more useful fine-grained features inside each divided patch, which are rarely considered**", the changed version is "However, **the rich spatial information inside these divided patches is rarely considered by previous models**".
>
> In Line97, the original description is "Han et al. propose Transformer iN Transformer (TNT) that **not only constructs** the global connection among outer patches, but also **the inner attention mechanism of each patch**", the changed version is "Han et al. propose Transformer iN Transformer (TNT) that **constructs not only** the global connection among outer patches, but also **the inner communication inside each patch**".
>
> In Line163, the original description is "Although the outer patches acquire fine-grained inner representations with original PyramidTNT **block**, the fusion method of simple flattening and addition ignores spatial information", the changed version is "Although the outer patches acquire fine-grained inner representations with original PyramidTNT **blocks**, the fusion method of simple flatten and add ignores spatial information".

---

### Official Review · Reviewer_oBDA · 2022-07-10

**Rating:** 7
**Confidence:** 5
**Soundness:** 3 good
**Presentation:** 3 good
**Contribution:** 3 good

**Summary:**

A fully transformer-based object detector (transformer-based backbone and neck) is an interesting topic. This paper proposes the method of the Fine-grained and Coarse-grained crossing representations for building efficient transformer-based object detectors. In the backbone, this paper maintains both the fine-grained and coarse-grained features and introduce a lightweight local-global cross fusion module. In the neck, the proposed module allows the detection tokens to make attention-based interaction with fine-grained representations firstly, and then perform further interaction with coarse-grained representations. The results on the COCO dataset demonstrate the effectiveness of the proposed method.

**Questions:**

1. Can the authors further explain the calculation process of Eq. (9)?

2. Where does the proposed efficient multi-scale feature integration demonstrate efficiency compared to other FPNs?

3. In which stage is the proposed multiscale feature integration used?


**Limitations:**

Yes, the authors have discussed the limitations and potential negative social impact of their work in the Appendix.

**Strengths And Weaknesses:**

Strengths
1. This paper is technically sound and easy to understand.

2. The proposed method is interesting and has a clear motivation. Other transformer-based detectors do not make good use of the fine-grained information, resulting in unsatisfactory performance. The proposed local-global cross fusion module, fine-coarse aware neck, and efficient multi-scale feature integration are novel.

3. The experiments are extensive and ablation studies are comprehensive to understand the proposed method.

Weaknesses

1. The writing needs to be improved slightly.

2. The results in Tab.1 are impressive, yet the authors are advised to make more discussions on the results to make the paper stronger.

---

> ### Author Response · Authors · 2022-08-02
> **Responses to Reviewer oBDA  (Part 4/4)**
>
> **Q2: Where does the proposed efficient multi-scale feature integration demonstrate efficiency compared to other FPNs?**
>
> **Q3: In which stage is the proposed multiscale feature integration used?**
>
> **A-Q2&A-Q3:** Thanks for the helpful comments. Since Q2 and Q3 are about Efficient Multi-scale Feature Integration (EMFI) , it is appropriate to answer these two questions together.
>
> Compared with other FPNs equipped with a large number of 3x3 convolution layers, our EMFI only consists of upsampling and 1x1 convolution, so it is relatively lightweight and brings little computational burden.
>
> EMFI is used in the backbone part. For backbones, such as ResNet, Swin-Transformer and PyramidTNT, we divide them into four stages, and each stage outputs feature maps with different scales. For a 2D image with scale of $H \times W $, with ResNet or Swin-Transformer as the backbone, the scale of feature maps in each stage is $\frac{H}{4} \times \frac{W}{4} $, $\frac{H}{8} \times \frac{W}{8} $, $\frac{H}{16} \times \frac{W}{16} $, $\frac{H}{32} \times \frac{W}{32} $. The feature map output by the first stage is ignored to put into Neck due to its insufficient feature extraction ability, and an additional feature map is generated by downsampling after the last stage. Therefore, taking ResNet or Swin-Transformer as the backbone, the scale of four output feature maps are $\frac{H}{8} \times \frac{W}{8} $, $\frac{H}{16} \times \frac{W}{16} $, $\frac{H}{32} \times \frac{W}{32} $,$\frac{H}{64} \times \frac{W}{64} $. These features are used in the Neck part to make cross-attention with det tokens. In our method, the scale of feature maps in each stage is $\frac{H}{8} \times \frac{W}{8} $, $\frac{H}{16} \times \frac{W}{16} $, $\frac{H}{32} \times \frac{W}{32} $,$\frac{H}{64} \times \frac{W}{64} $. In object detection, the feature map scale has a great impact on detection performance, so we keep the outputs shape of our method consistent with Swin-Transformer or ResNet.  Therefore, for the feature map with same scale of $\frac{H}{8} \times \frac{W}{8} $, the feature map obtained by Swin-Transformer is generated by 2 stages of Swin-Transformer blocks, while ours is only 1 stage of Transformer block. Compared with Swin-Transformer or ResNet, the feature representations of scale with $\frac{H}{8} \times \frac{W}{8} $ in PyramidTNT are relatively weak.
>
> Although the feature extraction capability of the first stage is limited, we still cannot ignore the output features of this stage. So we borrow the idea of FPN and the role of EMFI is to improve the feature representations of the first stage by introducing high-level features into low-level features. For other types of backbones, such as Swin-Transformer, EMFI is not necessary. We hope that this module brings as little computation as possible. Therefore, compared with other FPNs equipped with a large number of 3x3 convolution layers, our module only consists of upsampling and 1x1 convolution, which is relatively lightweight. Experimental results also prove that the AP is significantly improved by introducing EMFI with little increase in the amount of calculation.
>
>
> **Again, thank you very much for the kind efforts in evaluating and helping to improve the quality of our manuscript!**

---

> ### Author Response · Authors · 2022-08-02
> **Responses to Reviewer oBDA  (Part 3/4)**
>
>
> **Q1: Can the authors further explain the calculation process of Eq. (9)?**
>
> **A-Q1:** Thanks for the helpful comment. Eq. (7) and Eq. (9) are the Multi-Scale Deformable Attention between det tokens and global and local features respectively. Eq. (9) is the description of cross attention between det tokens and global features.
>
> The basic Multi-Head Attention in Transformer is as shown below:
>
>
> $$
> MultiHeadAttn(z_q,x_k)=\sum_{m=1}^{M} W_m[\sum_{k\in \Omega_k}^{} A_{mqk}\cdot W_{m}^{'} x_k]
> $$
> In the upper equation, $q \in \Omega_q$ indexes a query element with representation feature $z_q \in \mathbb{R}^C$, where $C$ is the feature dimension, $\Omega_q $ and $\Omega_k$ specify the set of query and key elements, respectively, $m$ indexes the attention head, $W_{m}^{'}\in \mathbb{R}^{C_v \times C}$ and $W_{m}\in \mathbb{R}^{C \times C_v}$ are of learnable weights ($C_v = C/M$ by default). The attention weights $A_{mqk} \propto exp{\frac{z_{q}^{T}  U_{m}^{T} V_m x_k}{\sqrt{C_v} } }$ are normalized as $ {\textstyle \sum_{k \in \Omega_k }A_{mqk}=1} $, in which $U_m, V_m \in \mathbb{R}^{C_v \times C}$ are also learnable weights.
>
> In Multi-Head Attention of Transformer, $k \in \Omega_k$ means that each query element must connect to all key elements, which causes a lot of computational burdens and makes the model difficult to train. In Multi-Scale Deformable Attention, each det token only needs to aggregate a small $K_O$ (usually set to 8) set of key contents sampled from the multi-scale feature maps $\mathcal{F}\_{O}^{l}$. The sampled values are automatically determined by the model in the training process. The equation is as follows:
> $$
> MSDA(Q_{det},\mathcal{F}\_{O}^{l})=\sum_{m=1}^{M_O} W_m \left [ \sum_{l=1}^{L} \sum_{k=1}^{K_O}
> A_{mlk}\cdot W_{m}^{'} \mathcal{F}\_{O}^{l}(\phi\_l(p)+\Delta p_{mlk})  \right ]
> $$
> where $Q\_{det}$ is the query element of one det token. $\phi_l(p)$  is the reference point of the det token re-scaled for the $l$-th level feature map, while $\Delta p\_{mlk}$ is the sampling offset for deformable attention. $\mathcal{F}\_{O}^{l}(\phi\_l(p)+\Delta p\_{mlk})$ represents the sampling values of the $l$-th feature map. In another word, MSDA only lets each det token interact with $K_O$ (usually set to 8) instead of all key elements of each feature map. This method greatly reduces the amount of calculation. MSDA also considers a multi-scale method that makes cross-attention with feature maps of different stages.

---

> ### Author Response · Authors · 2022-08-02
> **Responses to Reviewer oBDA  (Part 2/4)**
>
> **W2: The results in Tab.1 are impressive, yet the authors are advised to make more discussions on the results to make the paper stronger.**
>
> **A-W2:** Thanks for the helpful comment. The previous discussion about the results in Tab.1 is a little insufficient. We have made a more detailed illustration for this part, as shown below:
>
>   We conduct experiments under various computational constraints to demonstrate the effectiveness of our proposed FCDT. Specifically, Tab. 1 shows the comparison of FCDT with other state-of-the-art Transformer-based detectors, including DETR[11], SMCA[52], UP DETR[49], Efficient DETR[13], Conditional DETR[48], DAB DETR[27], DN DETR[50], SAM DETR[51], YOLOS[47], and ViDT[14] on COCO benchmark. The corresponding results elaborate the great potential in accuracy-computation trade-off of our proposed FCDT.
>
> **Compare with tiny detectors.** YOLOS[47] is a canonical ViT architecture for object detection. Although it has a small computational cost, the neck-free design withholds the YOLOS from obtaining high performance, our FCDT achieves +12.7 AP compared to the Deit-tiny based YOLOS. When compared to the recently proposed lightweight ViDT[14], our FCDT outperforms it by +2.6 AP with fewer FLOPs constraints. More specifically, the backbones of ViDT and FCDT attain similar results on ImageNet (74.9 of Swin-Nano v.s. 75.2 of P-Tiny), and the superiority in COCO further demonstrates the improvements brought by our proposed LGCF, FCAN, and EMFI.
>
> **Compare with small detectors.** We further compare our P-small based FCDT with Swin-Tiny based ViDT and several variants of ResNet-50 based DETR. For example, DN DETR[50] accelerates DETR training by introducing query denoising. Our FCDT outperforms it by +1.7 AP with far less computational cost (-17G FLOPs), and we still exceed the ViDT by +1.0 AP, and the amount of calculation is significantly reduced by 37G FLOPs.
>
> **Compare with medium detectors.** For the backbone with P-Medium, our FCDT achieves 48.1 AP with 173G FLOPs. In terms of AP, the detectors close to our method are DN DETR with DC5-ResNet-101 and ViDT with Swin-Small. Compared with them, the computational complexity of our method is lower than these detectors by 109G FLOPs and 35G FLOPs respectively due to our efficient detection framework. Besides, our method still reaches a better detection performance than theirs.
>
> In addition, when compared to those detectors with ResNet as the backbone, fully transformer-based models like ViDT and FCDT show better trade-off between accuracy and computational cost (higher AP and fewer FLOPs). This also reveals that fully transformer-based frameworks possess great potential for efficient object detection. And our proposed FCDT  can obtain the best trade-off among other detectors.

---

> ### Author Response · Authors · 2022-08-02
> **Responses to Reviewer oBDA  (Part 1/4)**
>
> **We would like to express our sincere thanks for the constructive comments and suggestions from the reviewer.**
>
>
>
> **W1: The writing needs to be improved slightly.**
>
> **A-W1:** Thanks for the helpful comment. We have checked the whole paper again, mainly to correct the typos and grammar errors. Some of these errors are modified as follows：
>
> In Line18, the original description is "**In the past decade,  models based on convolutional neural networks (CNNs) used to be the mainstream architecture for object detection tasks.**",  the changed version is "**The former mainstream architectures for object detection are mostly based on convolutional neural networks (CNNs).**"
>
> In Line25, the original description is "can be divided into three **parts**: backbone, neck and head",  the changed version is "can be divided into three **components**: backbone, neck, and head."
>
> In Line26, the original description is "With the development of transformer **used for** vision tasks", the changed version is "With the development of transformer **in** vision tasks".
>
> In Line27, the original description is "One is to **replace the backbone with transformer variants in CNN-based object detectors**", the changed version is "One is to **replace the CNN-based backbones with transformer variants in object detectors**".
>
> In Line38, the original description is "In DETR, ResNet is used as backbone for extracting features and transformer is proposed to integrate the relations between learnable object queries and extracted image features", the changed version is "In DETR, ResNet \cite{he2016deep} is used as **the backbone** for extracting features and transformer is proposed to integrate the relations between learnable object queries and **intermediate** image features".
>
> In Line46, the original description is "The other is the slow **convergence**", the changed version is "The other is the slow **convergence speed**".
>
> In Line61, the original description is "it incorporates an encoder-free neck structure to **boost** the detection performance without **much increase in computational load**", the changed version is "it incorporates an encoder-free neck structure to **further** **boost** the detection performance without **introducing too much computational burden**".
>
> In Line70, the original description is "**Considering the general object detection task**", the changed version is "**Take the general object detection benchmark as an example**".
>
> In Line91, the original description is "In this section, we briefly revisit the fine-grained representations in transformer and **detection transformer frameworks**", the changed version is "In this section, we briefly revisit the fine-grained representations in transformer and **transformer-based detection frameworks**".
>
> In Line137, the original description is "we introduce the lightweight bottom-up feature integration algorithm **Efficient Multi-scale Feature Integration module**", the changed version is "we introduce the lightweight bottom-up feature integration algorithm, **i.e., Efficient Multi-scale Feature Integration**".

---

### Meta-Review · Area_Chair_SmPG · 2022-08-27

**Recommendation:** Accept
**Confidence:** Certain

**Metareview:**

This work proposes a new object detector architector that is based on a CNN stem, combined with a mostly transformer-based architecture, with the addition of a cross-fusion module that allows for reconciling coarse and high-grained features for more precise object detection.

Thee paper is well-written, novel and presents a significant gain over the state of the art, using reasonable amount of compute.

Object detection is a central area of interest and this method shows how to leverage the power of transformers to push the envelope in this domain. Therefore, I propose this paper be accepted at NeurIPS 2022.

**Award:**

No

---

### Decision · Program_Chairs · 2022-09-14

Accept